Wound healing potential of mouth gel containing isopimarane diterpene from Kaempferia galanga rhizomes for treatment of oral stomatitis

Iadnut Anupon 1 2
http://orcid.org/0009-0006-8718-1626 Sae-lee Tanawan 1 2
Tewtrakul Supinya 1 2 3 supinya.t@psu.ac.th
1 Department of Pharmacognosy and Pharmaceutical Botany, Faculty of Pharmaceutical Sciences, Prince of Songkla University , Hat Yai, Songkhla , Thailand
2 Department of Chemistry and Center of Excellence for Innovation in Chemistry, Faculty of Science, Prince of Songkla University , Hat Yai, Songkhla , Thailand
3 Excellent Research Laboratory, Phytomedicine and Pharmaceutical Biotechnology Excellence Center, Faculty of Pharmaceutical Sciences, Prince of Songkla University , Hat Yai, Songkhla , Thailand
Iriti Marcello
Electronic publication date: 2024 Dec 18
Publication date: 2024
Volume: 12
Electronic Location ID: e18716
Received 2024 Sep 30; Accepted 2024 Nov 25
Copyright: © 2024 Iadnut et al.
Copyright year: 2024
Copyright holder: Iadnut et al.
License: This is an open access article distributed under the terms of the Creative Commons Attribution License, which permits unrestricted use, distribution, reproduction and adaptation in any medium and for any purpose provided that it is properly attributed. For attribution, the original author(s), title, publication source (PeerJ) and either DOI or URL of the article must be cited.
License URL: https://creativecommons.org/licenses/by/4.0/

Keywords: KG6, Wound healing, Kaempferia galanga, Mouth gel

Funding: Target-Based Research, Faculty of Pharmaceutical Sciences, Prince of Songkla University PHA6404015S Center of Excellence for Innovation in Chemistry PERCH-CIC Ministry of Higher Education, Science, Research, and Innovation This research work was supported by the Target-Based Research, Faculty of Pharmaceutical Sciences, Prince of Songkla University (PHA6404015S), which also supplied laboratory equipment, and the Center of Excellence for Innovation in Chemistry (PERCH-CIC), Ministry of Higher Education, Science, Research, and Innovation. The funders had no role in study design, data collection and analysis, decision to publish, or preparation of the manuscript.

==============================
Background

Oral ulcers have an impact on 25% of the global population including patients who are suffering from chemotherapy and radiotherapy treatments. Kaempferia galanga L. has been traditionally used for treatment of mouth sores and tongue blisters. However, the wound healing study of isopimarane diterpenes isolated from K. galanga is still limited.

Objective

This study aims to evaluate the wound healing potential of 6β-acetoxysandaracopimaradiene-1α,9α-diol (KG6), a compound isolated from Kaempferia galanga, by examining its biological activities. Additionally, we investigate the physicochemical and biological properties of (KG6) in formulated mouth gels.

Methods

The KG6 mouth gels at 0.10%, 0.25% and 0.50% w/w were formulated using sodium carboxymethylcellulose as a gelling agent, and their physicochemical and biological stabilities were assessed through a heating-cooling acceleration test. The quantification of KG6 contents in KG6 mouth gels was determined using gas chromatography. Both KG6 and KG6 mouth gels were evaluated for their wound healing properties including cell proliferation, cell migration, and antioxidant activity (H2O2-induced oxidative stress) in human gingival fibroblast (HGF-1-ATCC CRL-2014) (HGF-1). In addition, the anti-inflammatory activity against nitric oxide (NO) production was investigated in macrophage cells (RAW 264.7).

Results

After KG6 mouth gels were incubated under heating-cooling acceleration condition, the physicochemical properties of the KG6 mouth gels remain stable across various parameters, including appearance, color, smell, texture, pH, viscosity, separation, and KG6 content. The biological studies indicated that the KG6 compound possessed good wound healing potential. The 0.50% KG6 mouth gel exhibited marked anti-inflammatory effect by inhibiting NO production with an IC50 of 557.7 µg/ml, comparable to that of Khaolaor mouth gel, a positive control. The 0.25% KG6 mouth gel increased HGF-1 cell proliferation to 101.7–103.5%, whereas all formulations of KG6 mouth gel enhanced HGF-1 cell migration to 94.7–98.9%, higher than Khaolaor mouth gel (73.5%). Moreover, 0.50% KG6 mouth gel also showed a good antioxidant effect under H2O2-induced oxidative stress.

Conclusion

This study substantiates the significant biological activities related to the wound healing property of 0.50% KG6 mouth gel for treatment of aphthous ulcers and oral stomatitis from chemotherapy and radiotherapy treatments.

Introduction

Ulceration can be classified into many types based on the duration of ulcer, the number of lesions, and the location of the ulcer area (Minhas et al., 2019). One of the most prevalent areas where the ulcer manifests is the oral cavity. Oral mucosa consists of three layers including oral epithelium layer, lamina propria, and submucosa. Normally, oral ulcers appear from the oral epithelium layer to lamina propria (Waasdorp et al., 2021). Oral ulcers (oral stomatitis) affect worldwide up to 25% of the global population (Al-Maweri et al., 2023). Recurrent aphthous stomatitis (RAS), also known as canker sores, represents the most common disease of the oral membrane. Patients with RAS are unable to prevent it, and the clinical appearance of RAS is characterized by episodes of solitary or multiple painful ulceration without association with systemic diseases. The etiology and pathogenesis of RAS remain unclear. However, several factors are involved in RAS, including a positive family history, food hypersensitivity, smoking cessation, psychological stress, and immune disturbance (Akintoye & Greenberg, 2014; Gallo Cde, Mimura & Sugaya, 2009). RAS can be classified into three forms: minor (>70% cases), major (10%), and herpetiform (10%) (Edgar, Saleh & Miller, 2017). Those forms of RAS significantly affect the quality of life and interfere with daily life behavior. Moreover, oral stomatitis is also a common side effect that occurs in patients who receive chemotherapy and radiotherapy treatments.

Chemotherapy and radiation therapy are the most widely used for the treatment of cancers. Approximately 40% of patients undergoing chemotherapy develop oral mucositis. The risk of developing mucositis increases with the number of chemotherapy cycles and previous episode of chemotherapy-induced mucositis. The pathogenesis of chemotherapy induces mucositis was not fully elucidated, but it is thought to have two mechanisms: direct and indirect mechanism, caused by chemotherapy and radiation. Direct mucositis, the epithelial cells of the oral cavity are rapidly turnover, usually 7 to 14 days. The chemotherapy and radiation affect the maturation and cellular growth of epithelial cells, causing changes to normal turnover and cell death. The indirect mucositis can be caused by microbial infections from the suppression of immune system by chemotherapy and radioactive treatment (Naidu et al., 2004).

Wound healing is a necessary physiological process for the recovery of wounds resulting from ulcers. This process is composed of four stages, including hemostasis stage, inflammatory stage, proliferative stage, and the remodeling stage (Almadani et al., 2021). Hemostasis stage: when a wound occurs, platelets migrate to the wound area and stop bleeding of the blood. Moreover, platelets release chemokines to attract immune cells to the affected area (Waasdorp et al., 2021). Inflammatory stage: following the release of chemokines by platelets, the first immune cells that enter the wound area are neutrophils. Neutrophils undertake the clearance of dying cells and microorganisms around the wound by phagocytosis. In addition, neutrophils release chemokine and proinflammatory cytokine to activate other immune cells, such as lymphocytes and macrophages. Macrophages secrete proinflammatory cytokine and release growth factors to stimulate tissue regeneration (Mamun et al., 2024). Proliferation stage: keratinocytes, endothelial cells, and fibroblasts are proliferated and differentiated by receiving growth factors from macrophages (Gonzalez et al., 2016). Remodeling stage: myofibroblasts, derived from fibroblasts, release collagens to remodel tissue around the wound. At the end of the wound healing process, the wound from the ulcer will be cured (Atala et al., 2010). However, in some cases, wound healing is delayed by the excessive release of inflammatory mediators and reactive oxygen species (ROS) from macrophages (Wang et al., 2023). An alternative drug with the ability to increase wound healing and decrease excessive ROS production is of considerable interest.

The strategy to treat the oral ulcers is to enhance the wound healing process. Clinically, the drug used to treat oral ulcers is triamcinolone acetonide (TA) oral paste. TA oral paste is effective in the treatment of mucosal defects, and is used to relief the signs and symptoms of many oral ulcers from inflammatory conditions (Hamishehkar et al., 2015). However, TA is a member of steroid drug group which shows many side effects such as epidermal thinning, melanocyte inhibition, increases susceptible bacterial infection, allergic or irritant, perioral dermatitis, steroid addiction, and others (Coondoo et al., 2014). To avoid these side effects from TA oral paste, alternative natural product drugs are of interest.

Kaempferia galanga L., a medicinal plant belonging to the Zingiberaceae family, has been traditionally used as both a food and medicinal herb. The leaves and rhizomes of K. galanga are consumed as raw vegetables, cooked, or used as a spicy ingredient (Xiang, Ping & Tu, 2021). Furthermore, K. galanga rhizome has been utilized in the treatment of many diseases, including indigestion, ear inflammation, stomach pain, gastroenteritis, menstrual pain, and intestinal wounds (Yao et al., 2018). Decoction or powder derived from the K. galanga rhizomes is also applied for the treatment of numerous symptoms, such as mouth sores, tongue blisters, cold, cough, sore throat, asthma, pectoral, headache, toothache, dyspepsia, abdominal pain, diarrhea, and postpartum care (Kumar, 2020). The K. galanga rhizomes have been traditionally used in Thailand as the main constituent of topical application for treatment of aphthous ulcers more than 80 years (Aroonrerk & Kamkaen, 2018; Mekseepralard, Kamkaen & Wilkinson, 2010). Several studies demonstrated that bioactive compounds derived from K. galanga exhibited various pharmacological activities, including anti-inflammatory, anti-cancer, antimicrobial, and wound healing properties (Kochuthressia, Britto & Raphael, 2012; Kirana et al., 2002; Umar et al., 2014; Wahyuni et al., 2022). Thus, K. galanga represents a promising candidate for a new alternative drug that could improve the treatment of oral ulcers. However, the literatures on isopimarane diterpenes and gel containing isopimarane diterpenes from K. galanga in wound healing activity is still limited. 6β-Acetoxysandaracopimaradiene-1α,9α-diol (KG6), an isopimarane diterpene derived from K. galanga, has been reported to exhibit good anti-inflammatory activity (Tungcharoen et al., 2020). Therefore, this study aims to evaluate the wound healing potential of the KG6 compound and KG6 mouth gel by testing their biological activities on human gingival fibroblasts (HGF-1) and RAW264.7 cells. Additionally, the mucoadhesive KG6 mouth gel was tested for its biological, physical, and chemical stability using heating-cooling acceleration tests, making it a promising, stable, marketable product and an alternative therapeutic treatment for oral ulcers.

Methods

Reagents and chemicals

Dulbecco’s modified eagle medium (DMEM), Roswell Park Memorial Institute (RPMI) 1,640 medium, fetal bovine serum (FBS), 0.25% trypsin-EDTA, penicillin-streptomycin, phosphate buffered saline (PBS), and 3-(4,5-dimethylthiazol-2-yl)-2,5-diphenyl tetrazolium bromide (MTT) were purchased from Gibco® (Life Technologies, Paisley, Scotland). Dimethyl sulfoxide (DMSO), glycerin, sodium carboxymethylcellulose (SCMC), hydrogen peroxide (H2O2), sodium benzoate, indomethacin and allantoin were purchased from Sigma-Aldrich (Sigma-Aldrich, St. Louis, MO, USA). Khaolaor mouth gel was purchased from Khaolaor Laboratories (Khaolaor Laboratories Co., Ltd., Samutprakarn, Thailand). TA oral paste was purchased from Nida Pharma Incorporation (Nida Pharma, Ayutthaya, Thailand).

Plant material and isolated KG6 compound from K. galanga rhizomes

Rhizomes of K. galanga were provided from Prachinburi Province, Thailand, in November 2011 and were subsequently identified by Prof. Dr. Chayan Picheansoonthon, from the Royal Society of Thailand, Bangkok, Thailand. The voucher specimen was SKP 201110701 and was kept at the Department of Pharmacognosy and Pharmaceutical Botany, Faculty of Pharmaceutical Sciences, Prince of Songkla University, Thailand. The dried rhizomes, weighing 3 kg, were finely ground and extracted with 95% ethanol using a reflux condenser at 70 °C, repeated three times for 1 h each. The ethanolic extract powder obtained from K. galanga rhizomes weighed 282.2 g, with a yield of 9.73% w/w. This extract was subsequently fractionated according to bioassay-guided isolation methods as described previously (Tungcharoen et al., 2020). Initially, the ethanolic extract was fractionated into hexane, chloroform, ethyl acetate, and water fractions. The hexane fraction, which exhibited significant anti-inflammatory activity, was subsequently further purified using silica gel column chromatography, employing a gradient of hexane and ethyl acetate polarity to yield seven fractions. After that, fraction 3 was isolated and purified on silica gel column chromatography using hexane: ethyl acetate (9:1 and 7:3) and 100% chloroform as eluents, resulting in 12 subfractions. From these, subfraction 10 was recrystallized to yield KG6 compound.

Cell cultures and stock sample preparations

The human gingival fibroblasts (HGF-1-ATCC CRL-2014) (HGF-1) were obtained from the American Type Culture Collection (ATCC) and was cultured in a completed DMEM (DMEM supplemented with 10% FBS and 1% penicillin-streptomycin). RAW 264.7 cells (mouse macrophage-like cell line) were cultured in a completed RPMI 1640 (RPMI 1640 supplemented with 10% FBS and 1% penicillin-streptomycin). These cell lines were incubated at 37 °C in a humidified atmosphere in a 5% CO2 incubator. The stock solutions of pure compounds (i.e., KG6, allantoin, indomethacin and gallic acid) and gels (i.e., KG6 mouth gels, TA oral paste and Khaolaor mouth gel) were dissolved in DMSO and adjusted to 100 mM and 100 mg/ml, respectively. All stock solutions were further diluted to various concentrations with the culture media before the assays.

Formulation and characterization of mouth gel containing KG6 (KG6 mouth gel)

Formulation of KG6 mouth gels and heating-cooling acceleration test

SCMC was used as a gelling agent in this experiment. Six formulas of gel base (5.5–8.0% w/w SCMC) were formulated and characterized for appropriate physical properties (Table 1). Consequently, the optimal concentration of SCMC in the formulas was selected to create three different formulations of KG6 mouth gel (0.10%, 0.25%, and 0.50% w/w). The dispersion method was used to prepare the gel, whereas SCMC and sodium benzoate were dispersed in purified water and subsequently mixed gently using a mortar and pestle to form the gel base. After that, the active compound (KG6) was dissolved in glycerin and added to the gel base to prepare KG6 mouth gels. The components of both the gel base and KG6 mouth gels are recorded in Table 1. Once the KG6 mouth gels were successfully formulated, 30 grams of KG6 mouth gels were weighed and tested for physicochemical and biological stabilities by using heating-cooling acceleration test. This test evaluates the stability of formulated gel by mimicking extreme temperatures variations over a short period. It is primarily used to assess physical and chemical stability, predict shelf life, and ensure product safety and performance (Dantas et al., 2016). In this study, the test was conducted over six cycles for 12 days, with each cycle consisting of heating at 45 °C for 24 h followed by cooling at 4 °C for 24 h (Srirod & Tewtrakul, 2019).

Table 1 The components of gel base and KG6 mouth gel.

Chemical	Gel base (g)	Gel containing active compound (100 g)	
0.10%	0.25%	0.50%	
1. SCMC	5.50	6.00	6.50	7.00	7.50	8.00	8.00	8.00	8.00	
2. Sodium benzoate	1.00	1.00	1.00	1.00	1.00	1.00	1.00	1.00	1.00	
3. Glycerin	5.00	5.00	5.00	5.00	5.00	5.00	5.00	5.00	5.00	
4. Active compound (KG6)	–	–	–	–	–	–	0.10	0.25	0.50	
5. Purified water qs. to	100.00	100.00	100.00	100.00	100.00	100.00	100.00	100.00	100.00	

Physical characterization of KG6 mouth gels

Macroscopic organoleptic characteristics of KG6 mouth gels were inspected for their appearance, color, smell, and texture. The pH of KG6 mouth gels was measured using a digital pH meter (Thermo Scientific Co., Ltd., Waltham, MA, USA) at room temperature. The recorded pH values represented the average of three determinations. After that, viscosity measurements of KG6 mouth gels were conducted using a Brookfield Dial Reading Viscometer (Brookfield Engineering Laboratories Inc., Middleboro, MA, USA) at room temperature with a spindle number F (W&J Instrument Co., Ltd., Wujin, China) at a rotational speed of 10 rotations per minute (min). The measurement was performed in triplicate.

Chemical characterization of KG6 mouth gels

The content of KG6 in KG6 mouth gels was evaluated using gas chromatography (GC). The GC analysis of diterpenoids was previously described (Thanasakdecha & Tewtrakul, 2021). In brief, gels were dissolved in methanol, adjusted to a concentration of 10 mg/ml, and filtered through 0.45 µm nylon filters. Subsequently, 1 µl of the filtrated gel solutions were injected into an Agilent Technologies 7890 (Agilent Technologies Co., Ltd., Santa Clara, CA, USA) equipped with a HP-5 capillary column (30 m × 0.32 nm × 0.25 μm) in splitless mode. The analytical system was carried out under a helium flux of 1 ml/min. The column temperature was initially at 60 °C for 2 min, increased to 250 °C at a rate of 15 °C/min and finally increased at 5 °C/min until 305 °C. To quantify the content of KG6 in KG6 mouth gels, the standard calibration curve of KG6 was constructed. KG6 powder was weighted and dissolved in methanol, and the resulting solutions were filtered through 0.45 µm nylon filters. One microliter of the filtrated KG6 solution, at concentrations of 6.25, 12.50, 25.00, 50.00 and 100.00 μg/ml, was injected into the GC system. The amounts of KG6 in KG6 mouth gels were calculated using the linear regression equation obtained from the standard calibration curve of KG6.

Biological tests of KG6 compound and mouth gel containing KG6 (KG6 mouth gels)

Evaluation of anti-inflammatory property using RAW 264.7 cells

The inhibitory effect of KG6 and KG6 mouth gels on nitric oxide (NO) production in mouse macrophage-like cell line (RAW 264.7 cells) was evaluated by using the Griess reagent method (Sudsai, Tungcharoen & Tewtrakul, 2022). RAW 264.7 cells (1 × 105 cells/well) were maintained in complete RPMI-1640 medium and incubated at 37 °C with 5% CO2 for 1 h. After incubation, RAW 264.7 cells were induced by fresh RPMI-1640 medium containing 100 ng/ml of lipopolysaccharide (LPS). The LPS-induced RAW 264.7 cells were co-incubated with various concentrations of KG6, KG6 mouth gels, and positive controls (indomethacin was the positive control for KG6, while TA oral paste and Khaolaor mouth gel were positive controls for KG6 mouth gels) at 37 °C with 5% CO2 for 24 h. One hundred microliters of supernatant were aspirated from each well and reacted with 100 µl of Griess reagent. The optical density (OD) was measured with a microplate reader at 570 nanometer (nm) and the percentage of inhibition on NO production was calculated using the equation below. The half-maximal inhibitory concentration (IC50) was determined through the application of a logarithmic equation graph.

%Inhibition=[(Acontrol−Asample)/Acontrol]×100

Acontrol=Absorbanceofcontrol−Absorbanceofcontrolblank

Asample=Absorbanceofsample−Absorbanceofsampleblank

Evaluation of cell proliferation using HGF-1 cells

The measurement of cell proliferation was performed by using HGF-1 cells. HGF-1 cells (1 × 104 cells/well) were maintained in complete DMEM for 24 h. After incubation, HGF-1 cells were exposed to various concentrations of KG6, KG6 mouth gel, and positive controls. Allantoin served as the positive control for KG6, as it is a well-known compound recognized for its ability to enhance cell proliferation (Kim, Shin & Kim, 2018). Additionally, TA oral paste and Khaolaor mouth gel were used as positive controls for the KG6 mouth gel. The exposure took place at 37 °C with 5% CO2 for 48 h. Subsequently, 100 µl of supernatant were removed and 10 µl of 5 mg/ml MTT solution was added directly to the medium in each well, and the plate was then incubated at 37 °C for 2 h. After the incubation, all medium was removed and replaced with DMSO to dissolve the formazan crystal and the OD at 570 nm was recorded. The percentage of cell proliferation was calculated and compared to the non-treatment group (Sudsai, Tungcharoen & Tewtrakul, 2022).

% Cellviability= (Absorbanceoftreatmentgroup/Absorbanceofnon−treatmentgroup)×100

Evaluation of cell migration using HGF-1 cells

The migration of HGF-1 cells was examined by using a tip scratch assay. Briefly, HGF-1 cells (5 × 104 cells/well) were seeded into a 24-well plate and incubated at 37 °C with 5% CO2. After the confluent monolayer of HGF-1 cells was formed, cells were vertically scratched by using a 1,000 µl sterile pipette tip. Subsequently, cell debris was removed by washing with incomplete DMEM and 1 ml of fresh DMEM containing 2% FBS with or without samples and positive controls was added (allantoin was a positive control for KG6; TA oral paste and Khaolaor mouth gel were positive controls for KG6 mouth gels). Photographs were taken immediately and noted as day 0 by using an Eclipse TS100 inverted microscope (Nikon Co., Ltd., Shinagawa, Tokyo, Japan). The HGF-1 cells were incubated at 37 °C, and photographs were taken 3 times every 24 h, noted as days 1, 2, and 3, respectively. The migration of HGF-1 cells was assessed by measuring the length between the scratched cells (LBSC), and the analysis was performed using NIS-element D4 software. The percentage of cell migration was calculated and compared to the length obtained from day 0 (Sudsai et al., 2013).

% Cellmigrationofdayx=[(LBSCofday0−LBSCofdayx)/LBSCofday0]×100

wherex=1,2,and3

Evaluation of antioxidant activity using HGF-1 cells

H2O2-induced oxidative stress was used as a model to evaluate the antioxidant activity of KG6 and KG6 mouth gels. The methodology outlined by Zgorzynska et al. (2015) was utilized with a slight modification. In brief, HGF-1 cells (5 × 103 cells/well) were seeded in a 96-well plate and incubated at 37 °C with 5% CO2 for 24 h. After the time of incubation, the cells were treated with various concentrations of KG6, KG6 mouth gels, and positive controls for 24 h. Gallic acid was a positive control for KG6, while TA oral paste and Khaolaor mouth gel were used as positive controls for KG6 mouth gels. Subsequently, after pre-treatment with compounds or gels, the test wells were exposed to 3.75 mM H2O2 and incubated at 37 °C with 5% CO2 for 1 h. At the end of incubation, cell viability was assessed using the MTT assay.

Statistical analysis

Statistical analyses were conducted using the statistical package for the social sciences (SPSS) software. All data values were expressed as the mean ± SD based on three determinations. Statistical differences between groups were evaluated using a paired t-test and a one-way ANOVA, followed by Dunnett’s test. A significant level of p-values < 0.05 was considered to be statistically significant.

Results

Formulation of KG6 mouth gels

Physical stability of KG6 mouth gels

Before adding KG6 to KG6 mouth gels, the optimal concentration of SCMC should be verified. Six concentrations of SCMC were formulated into six gel bases (5.5–8.0% w/w SCMC), which were subsequently evaluated for their physical stability using a heating-cooling acceleration test (Table 1). The optimal formulation at 8.0% w/w SCMC demonstrated appropriate physical stability with a pH at 5.86 and a viscosity at 5.00 × 104 centipoise (cP). Consequently, the 8.0% w/w SCMC gel base was then selected to prepare three different formulations of KG6 mouth gels at 0.10%, 0.25%, and 0.50% w/w. Following the successful formulation of KG6 mouth gels, a heating-cooling test was conducted to assess their physical, chemical and biological stability properties. The results indicated that the appearance, color, smell, and texture remained unchanged before and after subjecting the gel to heating-cooling conditions. The pH and viscosity of the gel base and KG6 mouth gels were 5.86–5.89 and 5.00–6.67 (×104 cP), respectively. Notably, a significant increase in viscosity was observed in the gel base, 0.25%, and 0.50% KG6 mouth gels. The detailed results of the physical stability test are presented in Table 2.

Table 2 Physical stability of gel base and KG6 mouth gel before and after heating-cooling test.

Parameters	Gel base	0.10% KG6 mouth gel	0.25% KG6 mouth gel	0.50% KG6 mouth gel	
Before	After	Before	After	Before	After	Before	After	
Appearance	Semi-solid	Semi-solid	Semi-solid	Semi-solid	Semi-solid	Semi-solid	Semi-solid	Semi-solid	
Color	Colorless	Colorless	Milky white	Milky white	Milky white	Milky white	Milky white	Milky white	
Smell	Non scent	Non scent	Non scent	Non scent	Non scent	Non scent	Non scent	Non scent	
Texture	Homogeneous and smooth	Homogeneous and smooth	Homogeneous and smooth	Homogeneous and smooth	Homogeneous and smooth	Homogeneous and smooth	Homogeneous and smooth	Homogeneous and smooth	
pH values	5.86 ± 0.01	5.87 ± 0.00	5.88 ± 0.01	5.87 ± 0.02	5.85 ± 0.01	5.88 ± 0.01	5.89 ± 0.01	5.82 ± 0.02	
Viscosity (×104 cP)	5.00 ± 0.00	6.50 ± 0.50*	6.33 ± 0.29	6.67 ± 0.76	5.16 ± 1.04	6.00 ± 0.50*	5.03 ± 0.50	5.83 ± 0.29*	
Separation	No	No	No	No	No	No	No	No	
Notes:

Each value represents the mean ± SD of three determinations.

* Statistically significant difference between before and after heating-cooling test of samples, p < 0.05 (paired t-test).

Chemical stability of KG6 mouth gels

GC analysis was conducted both before and after heating-cooling accelerated condition. Quantification analysis of KG6 content in KG6 mouth gels with GC was calculated by comparing the retention time (min) and peak area (pA) with the KG6 standard calibration curve. The GC chromatograms of the KG6 compound and KG6 in the KG6 mouth gel, both with an average retention time of approximately 18 min (Fig. 1). The KG6 content in 0.10% KG6 mouth gel did not significantly change both before and after the heating-cooling test. However, the KG6 content of 0.50% KG6 mouth gel before the heating-cooling test (159.93 mg) was lower than that of after (167.61 mg) (Table 3).

Figure 1 GC chromatograms.

Gel base before heating-cooling (A), gel base after heatingcooling (B), 0.10% KG6 mouth gel before heating-cooling (C), 0.10% KG6 mouth gel after heating-cooling (D), 0.25% KG6 mouth gel before heating-cooling (E), 0.25% KG6 mouth gel after heating-cooling (F), 0.50% KG6 mouth gel before heating-cooling (G), 0.50% KG6 mouth gel after heating-cooling (H), and KG6 compound (I).

Table 3 KG6 contents obtained from GC analysis of mouth gel containing 0.10, 0.25, and 0.50% w/w of KG6 (6β-acetoxysandaracopimaradiene-1a,9a-diol) before and after heating-cooling test.

Sample	Amount of KG6 (6β-acetoxysandaracopimaradiene-1α,9α-diol)
(mean ± SD) (mg/30g)	
Before	After	
Gel base	Not detectable	Not detectable	
0.10% KG6 mouth gel	36.24 ± 2.12	35.62 ± 1.68	
0.25% KG6 mouth gel	77.86 ± 3.06	86.10 ± 3.65	
0.50% KG6 mouth gel	159.93 ± 2.18	167.61 ± 0.07*	
Notes:

Each value represents the mean ± SD of three determinations.

* Statistically significant difference between before and after heating-cooling test of samples, p < 0.05 (paired t-test).

Biological properties of KG6 and KG6 mouth gels

Isolated of KG6 compound

Approximately 171.9 g of ethanolic extract of K. galanga rhizome was fractionated into hexane (45.0 g, 26.2% w/w), chloroform (78.5 g, 45.7% w/w), ethyl acetate (0.98 g, 0.57% w/w), and water (47.4 g, 27.6% w/w) fractions. The hexane fraction, showing notable anti-inflammatory activity (IC50 = 8.5 µg/ml; 45 g), was further purified using silica gel column chromatography with a hexane and ethyl acetate polarity gradient, producing seven fractions. Fraction 3 underwent additional purification on silica gel using a hexane acetate mixture (ratios 9:1 and 7:3) and 100% chloroform as eluents, resulting in 12 subfractions. Subfraction 10 was recrystallized, yielding 1.49 g (0.87% w/w) of the KG6 compound, appearing as a white solid. Phytochemical characterization through 1H NMR, 13C NMR, and mass spectrometry identified KG6 as 6β-acetoxysandaracopimaradine-1α, 9α-diol, an isopimarane diterpene (Fig. 2).

Figure 2 Chemical structure of 6β-acetoxysandaracopimaradiene-1α,9α-diol (KG6).

Anti-inflammatory activity

The anti-inflammatory activity of KG6 and KG6 mouth gels was evaluated using RAW264.7 cells. The compound KG6 inhibited NO production with an IC50 of 36.3 µM, which was comparable to a positive control, indomethacin (IC50 = 38.2 µM), a nonsteroidal anti-inflammatory drug (Table 4). In addition, 0.10%, 0.25%, and 0.50% KG6 mouth gels, both before and after the heating-cooling test, exhibited good inhibition of NO production with IC50 values of 617.3–744.2, 599.6–679.9, and 557.7–575.6 µg/ml, respectively. Whereas gel base, both before and after the heating-cooling test, inhibited NO production with IC50 of 947.2–1,015.5 µg/ml (Table 5). Furthermore, the positive controls, TA oral paste and Khaolaor mouth gel, inhibited NO production with IC50 values of 225.9 and 500.3 µg/ml, respectively. The results indicated that the activity against NO production of 0.50% KG6 mouth gel (IC50 = 557.7 ± 84.4 µg/ml) was comparable to that of Khaolaor mouth gel (500.3 ± 87.2 µg/ml), but less than that of TA oral paste (225.9 ± 94.4 µg/ml). Furthermore, the anti-NO production of both gel base and KG6 mouth gels, before and after the heating-cooling test, did not show any significant difference (Table 5).

Table 4 Anti-NO production in RAW264.7 cells of KG6 from K. galanga.

Sample	% inhibition at various concentrations (µM)	IC50 (µM)	
0	1	3	10	30	100	
KG6	0.0 ± 0.0	10.0 ± 6.5	11.3 ± 4.9	19.2 ± 5.8	40.9 ± 2.6	77.9 ± 1.4	36.3 ± 4.0	
Indomethacin	0.0 ± 0.0	14.6 ± 8.0	15.2 ± 6.5	20.1 ± 8.6	43.3 ± 4.1	70.6 ± 17.3	38.2 ± 7.0	
Notes:

Each value represents the mean ± SD of three determinations.

No statistically significant differences were observed between control and various concentrations of sample (KG6 and indomethacin) at p < 0.05.

Table 5 Anti-NO production of gel base and mouth gel containing KG6 before and after accelerating conditions.

Sample	% Inhibition of various concentrations (µg/ml)	IC50 values (µg/ml)	
0	10	30	100	300	1,000	
Before	Gel base	0.0 ± 0.0	6.6 ± 3.6	9.5 ± 1.4	7.5 ± 5.5	14.5 ± 4.5	54.6 ± 11.3	1,015.5 ± 290.1	
0.10% KG6 mouth gel	0.0 ± 0.0	7.2 ± 9.4	8.1 ± 8.5	9.5 ± 7.0	14.4 ± 8.9	63.7 ± 16.5	744.2 ± 222.7	
0.25% KG6 mouth gel	0.0 ± 0.0	6.3 ± 0.6	7.3 ± 3.9	7.4 ± 4.7	9.5 ± 1.7	69.2 ± 11.1	679.9 ± 124.4	
0.50% KG6 mouth gel	0.0 ± 0.0	3.2 ± 3.0	7.6 ± 6.6	6.2 ± 6.0	14.2 ± 9.0	75.9 ± 6.4	557.7 ± 84.4	
After	Gel base	0.0 ± 0.0	1.5 ± 1.8	19.8 ± 14.9	25.9 ± 13.6	28.3 ± 17.1	54.2 ± 4.5	947.2 ± 425.7	
0.10% KG6 mouth gel	0.0 ± 0.0	20.3 ± 15.0	24.9 ± 12.8	23.0 ± 12.8	25.1 ± 17.7	64.4 ± 8.1	617.3 ± 128.7	
0.25% KG6 mouth gel	0.0 ± 0.0	9.6 ± 8.9	17.9 ± 16.1	22.2 ± 18.0	25.5 ± 20.4	65.1 ± 12.9	599.6 ± 62.7	
0.50% KG6 mouth gel	0.0 ± 0.0	11.7 ± 15.2	17.9 ± 11.8	20.7 ± 14.7	20.2 ± 14.1	69.4 ± 7.3	575.6 ± 94.6	
Positive control	TA oral paste	0.0 ± 0.0	19.6 ± 9.8	24.3 ± 12.9	24.4 ± 12.7	48.4 ± 15.7	91.1 ± 1.8	225.9 ± 94.4	
Khaolaor mouth gel	0.0 ± 0.0	−3.2 ± 4.7	−5.1 ± 2.0	−1.1 ± 1.1	13.8 ± 2.8	84.0 ± 7.8	500.3 ± 87.2	
Notes:

Each value represents the mean ± SD of three determinations.

No statistically significant differences were observed between control and various concentrations of sample (KG6 and Positive controls) at p < 0.05.

Evaluation of HGF-1 cell proliferation

The number of cells that proliferated from treated HGF-1 cells with KG6 and a positive control (allantoin) was investigated by the MTT assay. The result demonstrated that KG6 at the concentration of 30 µM significantly increased cell proliferation (114.9%) compared to both the control (100%) and allantoin (99.4%) (Table 6). Subsequently, all formulations of KG6 mouth gels both before and after heating-cooling test exhibited no toxicity towards HGF-1 cells, with the presented cell viability exceeding 90%. Furthermore, the 0.25% KG6 mouth gel both before and after heating-cooling test at concentrations of 10, 30, and 100 µg/ml exhibited a tendency to enhance the HGF-1 cell proliferation with the percentage of 100.4–101.3, 100.4–102.1, and 101.7–103.5%, respectively (Table 7). This suggested that the 0.25% KG6 mouth gel represented the optimal formulation for improving HGF-1 cell proliferation. Moreover, the cell proliferation effect of gel base and KG6 mouth gels was not significantly different between before and after heating-cooling test.

Table 6 Effect of KG6 and allantoin on HGF-1 cell proliferation.

Sample	% HGF-1 cell proliferation at various concentrations (µM)	
0	1	3	10	30	100	
Control	100.0 ± 0.0	–	–	–	–	–	
KG6	–	96.8 ± 2.2	98.8 ± 3.5	106.2 ± 3.4	114.9 ± 3.2a*	104.1 ± 4.4	
Allantoin	–	96.6 ± 8.9	99.7 ± 5.5	101.6 ± 3.6	99.4 ± 5.0	92.3 ± 5.3	
Notes:

Each value represents the mean ± SD of three determinations.

* Statistically significant difference between control and various concentrations of sample (KG6 and allantoin) at p < 0.05.

a >allantoin at p < 0.05.

Table 7 Effect of gel base and mouth gel containing KG6 before and after the accelerating conditions on HGF-1 cells proliferation.

Sample	% Viability of HGF-1 cells at various concentrations (µg/ml)	
0	1	3	10	30	100	
Before	Control	100.0 ± 0.0						
Gel base	–	97.1 ± 1.6	98.8 ± 0.6	98.7 ± 1.0	98.9 ± 1.2	99.4 ± 0.7	
0.10% KG6 mouth gel	–	91.9 ± 7.6	93.8 ± 8.4	95.9 ± 4.4	96.5 ± 3.2	96.6 ± 2.9	
0.25% KG6 mouth gel	–	99.1 ± 1.9	99.8 ± 2.2	101.3 ± 2.0	100.4 ± 3.1	101.7 ± 2.1	
0.50% KG6 mouth gel	–	94.1 ± 10.4	94.6 ± 9.1	96.5 ± 5.7	98.9 ± 2.0	99.6 ± 1.9	
After	Control	100.0 ± 0.0						
Gel base	–	96.0 ± 1.0	97.3 ± 1.0	98.6 ± 0.4	98.8 ± 1.4	98.7 ± 0.9	
0.10% KG6 mouth gel	–	97.2 ± 0.1	96.8 ± 0.7	96.0 ± 1.8	95.9 ± 1.7	96.0 ± 1.4	
0.25% KG6 mouth gel	–	98.8 ± 1.7	99.1 ± 0.6	100.4 ± 1.5	102.1 ± 3.6	103.5 ± 2.4	
0.50% KG6 mouth gel	–	99.4 ± 2.5	98.5 ± 0.5	99.6 ± 1.8	98.1 ± 1.6	97.2 ± 2.4	
Positive control	TA oral paste	–	101.6 ± 0.9	103.4 ± 8.8	97.5 ± 4.1	96.8 ± 1.4	98.6 ± 4.5	
Khaolaor mouth gel	–	104.3 ± 7.2	99.9 ± 6.2	89.0 ± 7.3	97.6 ± 14.8	90.2 ± 9.4	
Notes:

Each value represents the mean ± SD of three determinations.

No statistically significant differences were observed between control and various concentrations of sample (KG6 and Positive controls) at p < 0.05.

Evaluation of HGF-1 cell migration

The sterile tip was scratched vertically on the monolayer of HGF-1 cells. After that, HGF-1 cells were treated with samples and the length between the scratched cells was measured. The results in Fig. 3 and Table 8 demonstrated that KG6, at all concentrations including 10, 30, and 100 µM, significantly increased in the percentage of cell migration at 89.5, 88.8, and 91.8%, respectively, compared to the control (77.7%). Concordant with allantoin, there was a significant increase in the percentage of cell migration (94.4%) compared to the control (77.7%). These findings suggested that KG6 exhibited an effect on cell migration comparable to that of allantoin. KG6 was then incorporated into SCMC to form KG6 mouth gels at concentrations of 0.10, 0.25, and 0.50% w/w. The results in Table 9 demonstrated that all formulations of KG6 mouth gels including 0.10%, 0.25%, and 0.50%, exhibited a significant increase in the percentage of cell migration ranging from 96.3–98.9, 95.4–97.8, and 94.7–96.6%, respectively, compared to control (69.9%), TA oral paste (44.8%), and Khaolaor mouth gel (73.5%). TA oral paste and Khaolaor mouth gel, were employed as positive controls in the experiment. The result indicated that TA oral paste markedly decreased the cell migration (44.8%) while Khaolaor mouth gel mildly increased the cell migration (73.5%) when compared to the control (69.9%) (Table 9, Figs. 4 and 5). These findings suggested that KG6 mouth gels enhanced HGF-1 cell migration higher than both TA oral paste and Khaolaor mouth gel, which is a promising candidate for the treatment of oral ulcer. Furthermore, the results demonstrated that the percentage of the cell migration of gel base and KG6 mouth gels was not significantly different between before and after heating-cooling experiments, except for 0.10% KG6 mouth gel at 10 µg/ml; however, no impact on this difference was observed (96.3% of before and 98.9% of after) (Table 9).

Figure 3 The migration behavior of HGF-1 cells under the influence of the KG6 treatment.

Table 8 HGF-1 cells migration of compound KG6 and allantoin.

Sample	Dose (µM)	Length between the scratch (µm)	% Cell migration	
Day 0	Day 1	Day 2	Day 3	Day 1	Day 2	Day 3	
Control	–	992.2 ± 31.0	744.0 ± 20.7	302.3 ± 35.3	220.4 ± 16.4	24.9 ± 4.4	69.6 ± 3.0	77.7 ± 2.4	
KG6	10	1,000.7 ± 51.2	640.2 ± 12.0	355.2 ± 17.2	105.5 ± 28.1	35.9 ± 2.7	64.4 ± 3.6	89.5 ± 2.4*	
30	979.9 ± 21.9	684.5 ± 24.1	277.4 ± 154.0	109.2 ± 26.4	30.1 ± 3.4	71.7 ± 1.8	88.8 ± 2.7*	
100	992.2 ± 5.4	624.7 ± 5.0	334.9 ± 13.3	81.6 ± 24.7	37.0 ± 0.2	66.2 ± 1.5	91.8 ± 2.5*	
Allantoin	10	963.0 ± 6.4	584.7 ± 12.0	272.8 ± 70.9	80.8 ± 31.5	39.3 ± 0.8	71.6 ± 7.6	94.4 ± 5.9*	
Notes:

Each value represents the mean ± SD of three determinations.

* Statistically significant difference between control and various concentrations of each sample (KG6 and allantoin) at p < 0.05.

Table 9 HGF-1 cell migration of gel base and mouth gel containing KG6 before and after the accelerating conditions.

Sample	Dose (µg/ml)	Length between the scratch (µm)	% Cell migration	
Day 0	Day 1	Day 2	Day 3	Day 1	Day 2	Day 3	
	Control	–	1,034.7 ± 30.0	709.0 ± 80.0	403.5 ± 74.5	309.4 ± 9.4	31.5 ± 1.5	61.0 ± 10.0	69.9 ± 4.1	
Before	Gel base	10	1,089.2 ± 29.0	668.8 ± 115.3	413.2 ± 7.2	363.9 ± 33.4	38.7 ± 4.0	62.1 ± 1.7	66.7 ± 5.2	
100	1,063.2 ± 35.7	795.3 ± 115.3	443.1 ± 40.4	345.8 ± 115.9	25.2 ± 4.2	58.3 ± 4.2	67.2 ± 14.7	
0.10% KG6 mouth gel	10	1,172.6 ± 26.7	424.6 ± 44.5	237.5 ± 32.6	43.2 ± 4.5	63.8 ± 3.2	79.8 ± 2.4	96.3 ± 0.5abc*	
100	1,127.8 ± 22.7	432.6 ± 34.9	322.9 ± 27.4	30.3 ± 17.9	61.7 ± 2.6	71.3 ± 2.7	97.3 ± 1.6abc	
0.25% KG6 mouth gel	10	1,041.7 ± 68.4	342.2 ± 21.3	154.0 ± 20.4	48.2 ± 4.1	67.0 ± 3.9	85.2 ± 2.3	95.4 ± 0.5abc	
100	1,148.1 ± 99.6	439.3 ± 24.2	192.7 ± 45.8	37.9 ± 23.7	61.4 ± 6.9	83.2 ± 3.5	96.6 ± 2.1abc	
0.50% KG6 mouth gel	10	1,123.7 ± 17.5	394.2 ± 10.9	206.8 ± 11.3	60.0 ± 7.7	64.9 ± 0.9	81.6 ± 0.9	94.7 ± 0.6abc	
100	1,112.0 ± 30.5	366.1 ± 18.4	197.9 ± 8.7	37.4 ± 7.7	67.0 ± 2.0	82.2 ± 0.2	96.6 ± 0.9abc	
After	Gel base	10	1,088.5 ± 7.6	774.6 ± 14.0	546.2 ± 7.6	331.5 ± 35.1	28.8 ± 4.1	49.8 ± 2.1	69.5 ± 4.5	
100	933.5 ± 17.6	676.0 ± 25.0	453.4 ± 13.5	188.8 ± 13.5	27.5 ± 2.5	51.3 ± 6.8	79.7 ± 2.2	
0.10% KG6 mouth gel	10	1,239.5 ± 33.4	353.9 ± 8.2	162.9 ± 7.4	13.6 ± 3.5	71.4 ± 0.2	86.8 ± 0.8	98.9 ± 0.3abc*	
100	1,145.3 ± 46.7	422.4 ± 21.2	265.8 ± 8.1	28.6 ± 10.5	63.1 ± 1.6	76.8 ± 0.8	97.5 ± 0.9abc	
0.25% KG6 mouth gel	10	1,093.3 ± 44.8	387.8 ± 26.2	178.6 ± 20.8	49.5 ± 10.8	64.5 ± 3.5	83.6 ± 2.5	95.5 ± 0.9abc	
100	1,193.3 ± 31.3	386.7 ± 6.9	152.1 ± 22.4	25.7 ± 21.1	67.6 ± 1.4	87.2 ± 2.1	97.8 ± 1.8abc	
0.50% KG6 mouth gel	10	1,167.5 ± 72.9	423.6 ± 59.8	145.0 ± 19.4	46.0 ± 8.1	63.8 ± 3.1	87.6 ± 1.5	96.0 ± 0.8abc	
100	1,122.8 ± 70.9	316.4 ± 14.3	151.3 ± 18.2	40.6 ± 2.0	71.7 ± 2.7	86.5 ± 1.9	96.4 ± 0.2abc	
Positive control	TA oral paste	10	1,048.3 ± 37.0	844.0 ± 67.2	624.1 ± 52.1	576.7 ± 106.1	19.6 ± 4.4	40.3 ± 7.0	44.8 ± 11.5	
Khaolaor mouth gel	10	985.4 ± 23.5	637.0 ± 106.3	451.5 ± 121.7	261.4 ± 28.7	35.2 ± 11.5	54.1 ± 12.8	73.5 ± 2.9	
Notes:

Each value represents the mean ± SD of three determinations.

* Statistically significant difference between before and after the accelerating conditions, p < 0.05 (paired t-test).

Statistically significant difference between control and various concentrations of sample at p < 0.05. Dunnett’s test treated one group as a control and compared it with all other groups (a; > control, b; > TA oral paste, and c; > Khaolaor mouth gel).

Figure 4 The migration behavior of HGF-1 cells was examined in response to the before and after heating-cooling test of gel base and KG6 mouth gel treatment.

Figure 5 The migration behavior of HGF-1 cells was examined in response to treatment with the positive controls (TA oral paste and Khaolaor mouth gel).

Antioxidant activity

The antioxidant activity of KG6 and its formulated mouth gels was determined using an H2O2-induced oxidative stress assay, and cell viability was assessed through the MTT assay. The results showed that KG6 at all concentrations (1, 3, 10, 30, and 100 µM) slightly increased the percentage of HGF-1 cell survival in the range of 32.6–33.5%, compared to H2O2 treated group (30.3%), which only received H2O2 treatment. In addition, gallic acid (a positive control) at a concentration of 10 µM, exhibited a significant increase of HGF-1 cell survival with the percentage of 39.0% compared to the H2O2-treated group (30.3%) (Table 10). Subsequently, after the formulation of KG6 mouth gels, their antioxidant activity was evaluated. The result revealed that 0.50% KG6 mouth gel both before and after heating-cooling test, at concentrations of 30 and 100 µg/ml, significantly increased HGF-1 cell survival in the percentage of 50.2–53.2 and 50.4–53.8%, respectively, compared to the H2O2-treated group (35.8%) (Table 11). It is also indicated that 0.50% KG6 mouth gel (50.2–53.8%) exhibited an antioxidant effect higher than both TA oral paste (39.5–45.6%) and Khaolaor mouth gel (42.1–44.7%) when compared at the same concentrations (30 and 100 µg/ml). Moreover, the antioxidant activity of gel base and KG6 mouth gels was not significantly different between before and after heating-cooling test (Table 11).

Table 10 Effect of KG6 and gallic acid on H2O2-induced HGF-1 cell viability.

Sample	% Cell viability of HGF-1 cells at various concentrations (µM)	
0	1	3	10	30	100	
Control	100.0 ± 0.0						
H2O2	30.3 ± 1.6						
KG6		33.2 ± 7.4	33.3 ± 1.2	33.2 ± 4.3	32.6 ± 0.4	33.5 ± 0.6	
Gallic acid		33.5 ± 6.6	36.1 ± 4.9	39.0 ± 4.6*	37.9 ± 4.4	37.6 ± 4.7	
Notes:

Each value represents the mean ± SD of three determinations.

* Statistically significant difference between the H2O2-treated group (H2O2) and various concentrations of sample (KG6 and gallic acid) at p < 0.05.

Table 11 Effect of gel base and mouth gel containing KG6 on H2O2-induced HGF-1 cell viability.

Sample	% Cell viability of HGF-1 cells at various concentrations (µg/ml)	
0	1	3	10	30	100	
Control		100.0 ± 0.0						
H2O2		35.8 ± 2.9						
Before	Gel base	–	35.3 ± 3.6	36.0 ± 2.1	35.7 ± 2.3	34.6 ± 0.7	35.9 ± 0.7	
0.10% KG6 mouth gel	–	37.0 ± 0.7	37.0 ± 8.4	35.3 ± 1.1	33.8 ± 4.2	40.1 ± 2.2	
0.25% KG6 mouth gel	–	39.5 ± 7.3	39.9 ± 2.8	40.9 ± 3.4	38.4 ± 0.3	42.1 ± 0.6	
0.50% KG6 mouth gel	–	40.9 ± 3.1	39.1 ± 5.9	44.4 ± 8.1	53.2 ± 3.1a	53.8 ± 3.6a	
After	Gel base	–	36.0 ± 2.7	35.7 ± 1.8	36.7 ± 2.3	35.8 ± 1.0	35.1 ± 1.6	
0.10% KG6 mouth gel	–	36.0 ± 2.8	37.5 ± 0.5	34.7 ± 1.7	33.8 ± 0.7	38.8 ± 0.5	
0.25% KG6 mouth gel	–	38.5 ± 1.9	38.6 ± 3.0	40.4 ± 4.0	39.9 ± 4.5	44.5 ± 2.8	
0.50% KG6 mouth gel	–	40.5 ± 0.8	39.5 ± 3.7	44.9 ± 3.3	50.2 ± 4.8a	50.4 ± 13.6a	
Positive control	TA oral paste	–	29.3 ± 9.3	36.1 ± 7.1	33.1 ± 5.4	45.6 ± 3.3	39.5 ± 12.9	
Khaolaor mouth gel	–	43.3 ± 4.0	45.2 ± 2.6	43.0 ± 3.5	44.7 ± 2.1	42.1 ± 4.1	
Notes:

Each value represents the mean ± SD of three determinations.

Statistically significant difference between control and various concentrations of sample at p < 0.05. Dunnett’s test treated one group as a control and compared it with all other groups (a; > H2O2).

Discussion

As previously mentioned, oral ulcers represent one of the most prevalent medical conditions, affecting approximately 25% of the global population. RAS is a highly prevalent condition that impacts individual of all age groups, with a particular predilection for adolescents and young adults (Tarakji et al., 2015). The characterization of RAS is the recurrence of painful, ovoid or round shaped, single or multiple ulcers within the oral mucosa, and often affecting the non-keratinized mucosa (Bilodeau & Lalla, 2019). The severity of pain and discomfort associated with RAS can be serious, and significantly impact patients’ quality of life by interrupting routine oral functions such as eating, swallowing, and speaking (Rivera et al., 2022).

The drug delivery system designed for the oral cavity is referred as mucoadhesive dosage forms which facilitates intimate adherence between the oral mucus membranes and the dosage forms, thereby increasing the residence time, releasing the drug at the target site, and subsequently enhancing bioavailability. Mucoadhesive dosage forms are divided into four types: tablets, films, patches, and gel (Bhalerao & Shinde, 2013). Mucoadhesive gel is very interesting due to the use of mucoadhesive polymers such as SCMC, carbopol, hyaluronic acid, and xanthan gum. These polymers enhance viscosity, resulting in increased retention time as well as increased sustained and controlled release of the drug.

SCMC has been chosen to formulate mucoadhesive gel due to its appropriate properties including non-toxic, non-irritant to mucous membranes, high bioadhesion strength, odorlessness, tastelessness, and low cost (Al-Tayyar, Youssef & Al-hindi, 2020; Biswal & Singh, 2004). Furthermore, the formulation with SCMC did not require pH adjustment during the formulation process, whereas some gelling agents such as carbopol 934 required pH adjustment to a range of 5.5 to 7.0 using triethanolamine to form a gel. To avoid pH conditions that might interfere with the biological activity of KG6, we therefore did not use carbopol 934 as the gelling agent for this experiment. In addition, previous studies by Singh et al. (2015) demonstrated the successful formulation of ganglioside coated nanoparticles incorporated into SCMC gel base. This formulation exhibited a high degree of retention time, controlled localized infection, easily applied to patients with painless due to its soft smooth texture, prevented of new lesion forming, and exhibited the absence of observed side effects in periodontitis patients (Singh et al., 2015). Additionally, it has been reported that ten formulas of oral gel containing Punica granatum flower extract were evaluated for their physical properties. The results showed that formulations code F4 surpassed other formulations, exhibiting an optimal release pattern, proper appearance, stability, and the highest mucoadhesion ability. The enhanced mucoadhesion was attributed to the high content of mucoadhesive gelling agents, especially the main content of gelling agent, which was SCMC (Aslani, Zolfaghari & Davoodvandi, 2016).

In the present study, the different concentrations of SCMC ranging from 5.5% to 8.0% w/w were prepared in six formulations of gel bases and characterized for their physical properties. The formulation containing SCMC at a concentration of 8.0% w/w was selected for the preparation of KG6 mouth gels due to its superior viscosity range of 5.0–6.6 × 104 cP. This viscosity range resulted in a high degree of bioadhesion between the gel and mucous membranes. The pH of oral drug delivery should ideally fall within the range of 5.5 to 8.0, corresponding to the pH value of saliva. The formulation of KG6 mouth gels with SCMC results in a pH of approximately 5.8. This pH value prevents the disruption of the acid-base balance and homeostasis of the oral cavity organism (Maslii et al., 2020). The three concentrations of KG6 in the formulations 0.1%, 0.25%, and 0.5% w/w were selected for this study based on their reported wound healing activity, as documented by Sudsai, Tungcharoen & Tewtrakul (2022). Additionally, these concentrations have been used in studies of 2α-acetoxysandaracopimaradien-1α-ol (KM1), a compound isolated from Kaempferia marginata, which has also demonstrated wound healing activity (Thanasakdecha & Tewtrakul, 2021). The viscosity of 0.50% KG6 mouth gel exhibited a significant increase after heating-cooling test and the KG6 content in 0.50% KG6 mouth gel also increased after the heating-cooling test. These results may be due to the water evaporation from the formula. However, these values were still in acceptable ranges.

The anti-inflammatory activity of crude extracts from K. galanga has been previously reported. Lallo, Hardianti & Hayakawa (2022) conducted a screening test for anti-inflammatory activity using 35 ethanolic extracts of medicinal plants, revealing that the extract from K. galanga exhibited the most potent anti-inflammatory activity compared to all other plant extracts by inhibiting nuclear factor-kappa B (NF-κB) activity. Additionally, the essential oil from K. galanga rhizome was extracted through hydrodistillation and evaluated for its anti-inflammatory activity. The results demonstrated that the rhizome oil of K. galanga dose-dependently reduced NO production in LPS-induced macrophages (Chittasupho et al., 2022). Furthermore, earlier studies on isolated compounds from K. galanga, namely diarylheptonoids and ethyl-p-methoxycinnamate, have been reported for their anti-inflammatory activity by inhibiting NO production and reducing leukotriene B4 production, respectively (Dwita, Hikmawanti & Yeni, 2021; Yao et al., 2018). The present study focuses on the anti-inflammatory activity of an isopimarane diterpene, namely 6β-acetoxysandaracopimaradiene-1α,9α-diol (KG6), isolated from K. galanga. Tungcharoen et al. (2020) illustrated the mechanism of KG6 on anti-inflammatory activity, demonstrating its down-regulation of inflammation-related genes, including inducible nitric oxide synthase (iNOS), cyclooxygenase 2 (COX-2), and tumor necrosis factor-alpha (TNF-α) in the LPS-induced macrophages cell line (RAW 264.7 cell). Consequently, KG6 emerges as a promising candidate for treating inflammatory-related diseases and can potentially be formulated into a mucoadhesive dosage form for easy application to the target site. Since mucoadhesive gel is interesting for oral drug delivery system, the KG6 was then incorporated into the SCMC gel base. The anti-inflammatory activity of these mucoadhesive gels was evaluated. In the present study, we used two positive controls: TA oral paste and Khaolaor mouth gel. TA oral paste is a corticosteroid anti-inflammatory drug with 0.1% w/w triamcinolone acetonide, a standard treatment for oral ulcers. Khaolaor mouth gel is a marketable herbal product that alleviates symptoms of oral inflammation and aphthous ulcers. It contains Kaempferia galanga, peppermint oil, menthol, and other herbs (https://www.khaolaor.com/product/5966-6625/khaolaor; Khaolaor Laboratories Co., Ltd., Samutprakarn, Thailand). Peppermint essential oil, one of the active ingredients in Khaolaor mouth gel, has demonstrated anti-inflammatory and wound-healing properties by reducing inflammatory cytokines, enhancing cell migration, promoting collagen synthesis, and supporting the re-epithelialization of fibroblast cells (Modarresi, Farahpour & Baradaran, 2019). The present results showed that all formulations of KG6 mouth gels exhibited good anti-inflammatory activity, with no differences observed between the samples before and after the heating-cooling test. This finding suggests that these mouth gels demonstrated good stability in terms of their anti-inflammatory activity. Therefore, the KG6 mouth gels may serve as a promising candidate for treating oral ulcers by releasing the active compound (KG6) into the oral cavity’s ulcer area. Moreover, various extracts from K. galanga, such as aqueous (Sulaiman et al., 2008), alcoholic (Ridtitid et al., 2009; Vittalrao et al., 2011), chloroform, hexane and petroleum ether extracts (Jagadish et al., 2016) have been reported to exhibit anti-inflammatory activity in animal studies. Since KG6 is one of the main components in K. galanga rhizomes, these previous reports might therefore support the use of KG6 mouth gels as anti-inflammatory agents for wound treatment.

Cellular proliferation and migration are the important processes for the secondary phase of wound healing. These processes initiate within the first 48 h until the 14th day after the onset of the wound (Gonzalez et al., 2016). Moreover, they involve dynamic interactions and crosstalk among cells, interactions with molecules of the extracellular matrix, and the production of mediators and cytokines into the wound area to regenerate tissue (Grada et al., 2017). Previous investigations have explored the wound healing activity of K. galanga. Wahyuni et al. (2022) reported that the ethanol extract of K. galanga L. rhizome exhibited wound healing properties by increasing the percent recovery of the wound area in the rat oral mucosa. Another study by Tara Shanbhag et al. (2006) revealed the effective impact of the ethanolic extract of K. galanga on three types of wound models, including incision, dead space, and excision wound models. The results showed that the ethanolic extract of K. galanga increased the wound breaking strength in the incision wound, increased collagen maturation in the dead space wound, and enhanced epithelialization in the excision wound (Tara Shanbhag et al., 2006). However, an investigation of the wound healing activity of isolated compounds from K. galanga is necessary. Therefore, our study reports on the wound healing properties of KG6 isolated from K. galanga rhizome as well as KG6 mouth gels. The results demonstrated that KG6 increased HGF-1 cell proliferation and migration. Interestingly, 0.25% KG6 mouth gel increased HGF-1 cell proliferation, whereas all formulations of KG6 mouth gels significantly enhanced HGF-1 cell migration, better than both TA oral paste and Khaolaor mouth gel, which served as positive controls in the experiment. Notably, TA oral paste markedly decreased HGF-1 cell migration due to the corticosteroid triamcinolone acetonide, known for its inhibitory effect on cell migration (Wang et al., 2002). There were no significant differences in cell proliferation results before and after the heating-cooling tests. This was consistent with the cell migration results, except for the 0.10% KG6 mouth gel at a concentration of 10 µg/ml. This exception may have been due to randomly sampling the gel from a concentrated area in the container, which was caused by water evaporation in the 0.10% KG6 mouth gel following the heating-cooling test, resulting in a higher concentration of KG6. These results suggest that the KG6 mouth gel maintained its stability in enhancing both cell proliferation and cell migration. The effects of KG6 mouth gels on HGF-1 cell migration were consistent with findings by Sudsai, Tungcharoen & Tewtrakul (2022) where KG6 gels were evaluated for the wound healing activity using human dermal fibroblasts (HDF), the cell type mainly presents in skin connective tissue. They reported that KG6 gels slightly increased HDF cell proliferation and markedly increased HDF cell migration in all formulations (Sudsai, Tungcharoen & Tewtrakul, 2022). Thus, our finding suggests that the primary mechanism of KG6 mouth gels in the secondary phase of wound healing could mainly be the enhancement of HGF-1 cell migration.

Recently, we have learned that oxidative stress is a phenomenon that regulates wound healing processes when the content of ROS is optimal. The appropriate level of ROS is beneficial by preventing wound infection and promoting wound healing (DeCoursey, 2016; Rezende et al., 2018). The low concentration of H2O2 at the wound site has been proven to have a chemotaxis effect on neutrophil (Klyubin, Kirpichnikova & Gamaley, 1996). H2O2, the member of ROS, plays an important role in wound healing by disinfection, activating keratin-forming cell regeneration, recruiting neutrophils, and promoting angiogenesis (van der Vliet & Janssen-Heininger, 2014). However, the excessive ROS at the wound area results in wound healing impairment by reducing fibroblasts, keratinocytes, and endothelial cells (Sampson, Berger & Zenzmaier, 2012). Moreover, a high level of ROS reduced platelet adhesion capacity and aggregation number thereby the wound healing process could be delayed (Hosseini et al., 2020). Therefore, the antioxidant activity of natural products needs to be explored to control the ROS level into the non-toxic range in the case of excessive ROS in the wound area (Wang et al., 2023). Our study reveals the antioxidant activity of KG6, an isopimarane diterpene, which is a promising compound for controlling ROS level and enhancing the wound healing process. The results showed that the KG6 compound has moderate antioxidant activity. However, when incorporated into the KG6 mouth gels, the gel exhibited enhanced antioxidant activity, particularly in the 0.50% KG6 mouth gel at a concentration of 100 µg/ml (50.4–53.8%), which significantly increased the antioxidant effect compared to the control group (35.8%, H2O2-treated group). This suggests that combining KG6 with SCMC gel enhances antioxidant properties, which aligns with findings from Nabeel Ahmad et al. (2024), who demonstrated that incorporating pomegranate peel extract (PPE) into carboxymethyl cellulose increased antioxidant activity and water resistance. In addition, there are no differences between before and after heating-cooling tests in the results of antioxidants activity. This indicated that KG6 mouth gels represented good stability on their antioxidant activity under H2O2-induced oxidative stress. Thus, KG6 mouth gel could be an alternative choice to control the level of ROS in the wound area resulting in the enhancement of the wound healing process.

Conclusion

This study provides strong confirmation that KG6, an isopimarane diterpene isolated from K. galanga, exhibits significant wound healing activity. This effect is attributed to reducing inflammation, increasing gingival fibroblast cell proliferation and cell migration as well as attenuating excessive ROS under oxidative stress conditions. The KG6 mouth gel was successfully formulated using SCMC as a gelling agent. The gel formulation technique was modified to simplify the preparation process and avoid changing the pH, which can maintain the biological activity of the compound, demonstrating significant oral wound healing potential. KG6 mouth gels also demonstrate good physicochemical and biological stabilities. This study supports the traditional use of K. galanga for wound repair. Finally, the ultimate challenge goal is the prospective study in animal models, which is a promising approach for the treatment of oral ulcers in the future.

Supplemental Information

Supplemental Information 1 Raw data.

Additional Information and Declarations

Competing Interests

Author Contributions

Data Availability

The authors declare that they have no competing interests.

Anupon Iadnut conceived and designed the experiments, performed the experiments, analyzed the data, prepared figures and/or tables, authored or reviewed drafts of the article, and approved the final draft.

Tanawan Sae-lee performed the experiments, prepared figures and/or tables, and approved the final draft.

Supinya Tewtrakul conceived and designed the experiments, analyzed the data, authored or reviewed drafts of the article, and approved the final draft.

The following information was supplied regarding data availability:

The raw data is available in the Supplemental File.

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
