# Peer review of "Wound healing potential of mouth gel containing isopimarane diterpene from Kaempferia galanga rhizomes for treatment of oral stomatitis"

_PeerJ, doi:10.7717/peerj.18716_

## Round 0.1 · original submission · Major Revisions

Authors should emphasize the contribution of their study compared to previous study (see comment of reviewer 2).

Reviewer 1 ·

Basic reporting

Really I am so delighted to review the manuscript (#106471). This study has put good efforts for investigating the wound healing potential of a newly formulated mouth gel using human cell lines. Additionally, they investigate the bio- and physicochemical properties of this oral gel.

I suggest some parts of the article need more work.

1- Abstract section should be formatted according to the journal instruction: “Headings in structured abstracts should be added in bold and followed by a period”. Here, no headings are added. Add subheadings (Background, Objective, Methods, Results, and Conclusion). Although the main hypothesis to be tested is mentioned, it needs to be rewritten in clearer detailed way in both Abstract & Introduction sections.
Keywords: It will be better to add some relevant keywords to the existing ones.
2- Introduction section: Some sentences need reference to be added. Aim of the work needs to be rewritten in clearer detailed way in both Abstract & Introduction sections.
3- M & M section: Proper statistical analyses description needs to be clarified.
4- Results section: Some paragraphs do not belong to results and should be transferred to M & M section or to Discussion section. Many paragraphs need to be rephrased and clarified. Also result redundancy should be deleted.
5- Discussion section: Some paragraphs do not belong to results and should be transferred to M & M section or to Results section.
6- Conclusion section: Lack some important findings which must be added.
7- References section: Should be revised according to PeerJ format.
8- English language and style: English language of the whole MS is generally acceptable with few sentences need to be rephrased.
The overall workload is good but the specific hypothesis to be tested needs to be rewritten in clearer detailed way in both Abstract & Introduction sections.

In my opinion, all figures are presented well and some of which need little re-work.

Experimental design

The authors analyzed the wound healing potential of a newly formulated mouth gel using human cell lines. Additionally, they investigate the bio- and physicochemical properties of this oral gel.
Although the authors presented important findings, some parts of the manuscript need a further review.

1- The purpose of the study needs to be rewritten in clearer detailed way in both Abstract & Introduction sections.
2- In methods section, authors described statistical analyses done but don’t mention why they did every specific test. Few other methodologies need more details, and clarification.
Line by line review is provided in the annotated pdf file, you will find some suggestions.

Validity of the findings

1- In Results section, although the authors did statistical analyses, in some parts there is no reflection of these analyses in writing the results. PLZ present the results in the light of your statistical analyses.
2- Also, the results should be described as they are presented in Figs and Tables. You should describe your finding without redundancy and without speculations in Results section.
3- In addition some sentences in the Results belong to discussion section (Interpretations and speculations). Other paragraphs lack clear presentation and use very long sentences and hard to read (kindly rephrase). Other paragraphs belong to M & M section (those interpreting methods or why we did such analyses).
4- Finally, figures are well-constructed (See line by line review in the pdf file).
5- In Discussion section, the same comment on Results section. Some paragraphs lack clear presentation and use very long sentences and hard to read (kindly rephrase). Other paragraphs belong to other sections (Kindly put in their proper section). I included some specific comments in the pdf (See line by line review in the pdf file).
6- The Conclusion section needs addition of the important findings.

Additional comments

1- PLZ consider that the abstract in PeerJ format including subheadings (Background, Objective, Methods, Results, and Conclusion). Headings in structured abstracts should be added in bold and followed by a period. Additionally, make the subsequent required corrections.
The purpose of the study needs to be rewritten in clearer detailed way in both Abstract & Introduction sections.
2- In Introduction, Please add the required references and make the subsequent corrections.
3- In statistical analysis topic of the M & M section mention why every test was done?. Also as you have repeated your experiments only thrice, it is more accurate to use SD instead of SE.
4- In Results make the required corrections to your Figures and Tables.
Describe your text of results in the light of your well-presented Figures, and statistical analyses you did, without redundancy.
5- Figures 4 and 5 are better to be one figure A and B. for all Tables please explain the statistical analyses if present and ad if it is absent.
6- In Results section, just describe your findings without interpretation or speculations.
7- In discussion section, interpret your results, compare with the previous results in the subject (agree or disagree), give reasonable interpretation to the disagreeable research, and finally extract your definite conclusion.
8- References should be revised in accordance with the PeerJ format.
9- References in the text should be italicized in arranged in ascending years all over the MS as the journal formatting.
10- Extra-spacing should be revised all over the MS.
11- In the Results section, references should NOT be cited.

Annotated reviews are not available for download in order to protect the identity of reviewers who chose to remain anonymous.

Reviewer 2 ·

Basic reporting

This manuscript evaluates the oral wound healing potential of KG6 mouth gel. The manuscript is well organized, but remain some points to be improved.

General
1. I found a previous literature titled as “Anti-inflammatory and wound healing effects of mouth gel containing kaempulchraol K from Kaempferia galanga rhizomes. J Ethnopharmacol 2024;324:117762”. Based on the findings from the previous one, this manuscript does not add a significant level of knowledge to the existing literature.

Introduction
2. This manuscript evaluated wound healing effects of KG6 under heating-cooling acceleration. Why did the authors adopt this method? The benefits/purpose of heating-cooling in oral wound healing should be added to the introduction to clarify why heating-cooling was used as an accelerating condition.

3. Line 133: Modify to (KG6)

Experimental design

Materials & Methods
4. The purpose of this study was to assess the oral wound healing effect of the gel. I wonder why the RAW 264.7 cells were used for the anti-inflammatory test of the gel while the scratch wound test was performed with HGF cells. It would be better to use HGF cells for the anti-inflammatory test, in order to simulate the in vitro oral environment.

5. Please clarify the concentration of H2O2.

6. There is the inconsistency in the dose of KG6 among the in vitro test. Please refer to Table 6 and the others.

Validity of the findings

Results
7. Please add the peak of KG 6 in Figure 2 (B).
8. Regarding the figures 3-6, please provide figures with high-resolution. The differences are hardly discernible. Additional vertical lines would be beneficial to compare cell migration levels between groups.
9. Regarding the tables 5, 7, 9, and 11, please add the explanation of the % Inhibition of various concentrations (µg/ml). I cannot clearly understand why the various concentrations were applied to 0.1, 0.25, and 0.5% KG6 gel. I think that the effects will be affected by total amounts of KG6. Also, please revise the table adding the statistical annotations to improve readability. It is necessary to show statistical comparisons of each concentration. In addition, to demonstrate that the anti-inflammatory effect is due to KG6, a significant difference should be observed between the gel base and the KG6 mouth gel.
10. Line 364~365: What is your evidence for thinking that the 0.25% gel is the optimal formulation, given that nothing in Table 7 shows a significant difference from the control?
11. Line 377~378: While the cell migration effect of KG6 compared to the control is important, it is also important to look at the differences between the concentrations of KG6 and to compare KG6 to allantoin. In Table 8, all concentrations of KG6 show superior cell migration compared to the control. However, line 377 mentions that 100 µM shows comparable cell migration to allantoin, so the significance of allantoin and KG6 needs to be checked and added to Table 8.
12. Line 389: It would be good to state what the increase in cell migration at 10 µg/ml after heating-cooling experiments.
13. In Table 10, there is no difference between only H2O2 and KG6 treatment. But in Table 11, cell viability is increased in 0.5% KG6 mouth gel than in only H2O2 group. Therefore, the increase in cell viability in the 0.5% KG6 mouth gel is somewhat difficult to attribute to KG6 and I wonder if it is not due to the gelling agent, SCMC.
14. Table 11: The positive control should be a drug with proven efficacy. In the introduction, you state that TA oral paste is clinically used for the treatment of mouth ulcers. However, in Table 11, the positive control group is not significantly different from the H2O2 only group. I was wondering what your thoughts are on this.

Additional comments

Discussion
15. Line 507: Table 9 shows that all concentrations of KG6 mouth gel accelerated cell migration over the positive control, is there a reason only 0.25% KG6 mouth gel is mentioned?
16. Line 510: The introduction mentions that the strategy to treat oral ulcers is to promote the wound healing process and TA oral paste is used clinically. However, the discussion states that TA oral paste reduces HGF cell migration. The introduction and discussion are inconsistent. A positive control is a comparison of an experimental group to a drug with a known effect, so if TA reduces cell migration, as you wrote in the discussion, it can’t be used as a positive control.
17. Line 535: There is no indication of significance by concentration in Table 11. Therefore, it would be difficult to conclude that an antioxidant is present, especially at the concentration of 100 µg/ml in 0.5% KG6 mouth gel. The significance by concentration should be checked and the discussion revised.

Conclusion
18. The authors concluded that 0.5% KG6 mouth gel showed significant oral wound healing potential. Based on the conclusions made by the authors, I doubt why the authors treated the additional 1-100 µg/ml of gel with different dose of KG6.

---

## Round 0.2 · accepted · Accept

Now MS can be accepted, I confirm that the authors replied satisfactorily to both reviewers